# Assessment of glycemic control, health-related quality of life, and associated factors in type 2 diabetic patients attending a comprehensive specialized hospital in Northwest Ethiopia

Assefa Belay Asrie [1]*, Tafere Mulaw Belete[1], Melshew Fenta Misker[1], Alemante Tafese Beyna[1], Habtamu Semagne Ayele[1], Kidist Goshime Tekle[2], Yonas Zewdu Milikit[3], Ephrem Adane Andargie[4], Hiwot Tesfaselassie Afework[5], Yenatfanta Gezu Lenjiso[5], Gebrehiwot Lema Legese[6]

1 Department of Pharmacology, School of Pharmacy, College of Medicine and Health Sciences, University of Gondar, Gondar, Ethiopia, 2 Department of Pediatrics and Child Health, School of Medicine, College of Medicine and Health Sciences, University of Gondar, Gondar, Ethiopia, 3 Department of Ophthalmology, School of Medicine, College of Medicine and Health Sciences, University of Gondar, Gondar, Ethiopia, 4 Department of Gynecology and Obstetrics, School of Medicine, College of Medicine and Health Sciences, University of Gondar, Gondar, Ethiopia, 5 Department of Anesthesiology, Critical Care, and Pain Medicine, School of Medicine, College of Medicine and Health Sciences, University of Gondar, Gondar, Ethiopia, 6 Department of Internal Medicine, School of Medicine, College of Medicine and Health Sciences, University of Gondar, Gondar, Ethiopia

* assefabelay@gmail.com

## Abstract

### Background

Poor glycemic control in type 2 diabetes mellitus (T2DM) leads to serious complications that negatively impact health-related quality of life (HRQoL). This study aimed to assess glycemic control, HRQoL, and their associated factors in T2DM patients.

### Method

This is a cross-sectional study and was conducted from May 1 to July 30, 2024. Systematic random sampling technique was used to recruit the study participants. The average fasting blood glucose (FBG) levels of three consecutive tests during follow-up visits were extracted from patient medical records while the data pertaining HRQoL were collected through interview using EQ-5D five-level (EQ-5D-5L) questionnaire and EQ visual analog scale (EQ VAS). EQ-5D-5L utility scores were determined using disutility values established for Ethiopian context. The FBG level was used to categorize patients by glycemic status (controlled or uncontrolled). Binary logistic regression analysis was performed to outline factors associated with glycemic control. Mann–Whitney U and Kruskal–Wallis tests were used to compare the median utility and VAS scores between subgroups. Furthermore, Tobit regression analysis was performed to determine factors associated with HRQoL.

**Data availability statement:** All relevant data are within the manuscript and its Supporting Information files.

**Funding:** The author(s) received no specific funding for this work.

**Competing interests:** The authors have declared that no competing interests exist.

## Results

Nearly half (48.7%) of the patients were with uncontrolled glycemic levels (out of the target 4.4−72 mmol/L). In the multivariate logistic regression analysis, age, disease duration, comorbid conditions, diabetes complications, adherence to antidiabetic medications, and herbal medicine use were associated with glycemic control. Pain/discomfort, performing usual activities, and anxiety/depression were HRQoL dimensions in which the majority of participants reported problems; 85.8%, 76.2%, and 74.6% of participants, respectively, reported having problems in the dimensions. The overall median (interquartile range) EQ-5D-5L utility score was 0.86 (0.76–0.93) while EQ VAS score was 75.0 (65.0–80.0). The Tobit regression analysis showed that older age, diabetes duration, comorbid conditions, diabetic complications, and herbal medicine use were significantly negatively associated with HRQoL scores. On the other hand, engagement in physical exercise, controlled glycemic level, and adherence to antidiabetic treatments were found to be positively associated.

## Conclusion

In conclusion, nearly half of the patients were with uncontrolled glycemic level. The majority of participants reported problems in pain/discomfort, usual activities, and anxiety/depression dimensions of HRQoL. Several factors were correlated with both glycemic control and HRQoL. Adherence to antidiabetic medications was positively associated with both glycemic control and HRQoL. In contrast, older age, longer duration of diabetes, presence of comorbidities, diabetic complications, and use of herbal medicine were all negatively associated with both outcomes. On the other hand, adherence to dietary recommendations was positively associated only with glycemic control, while engagement in physical exercise was positively associated only with HRQoL. Moreover, glycemic control was associated with improved HRQoL. The findings underscore the importance of interventions targeting modifiable factors, such as dietary modifications, physical activity, and adherence support, to improve overall glycemic control and HRQoL.

## Background

Diabetes is a disease that results from the failure of the pancreas to produce adequate insulin or when the body is unable to utilize the insulin efficiently [1]. It is commonly characterized by hyperglycemia and it may cause serious damage to various systems of the body, especially the nerves and blood vessels. Type 1, type 2, and gestational diabetes mellitus are the most commonly known forms, though there are also other specific and much less common types [2]. Type 2 diabetes (T2DM) is the most prevalent of them, accounting for 90–95% of diabetes cases [3]. The prevalence of diabetes was 9.3% worldwide in 2019 and that number is estimated to increase to 10.2% by 2030 and 10.9% by 2045 [4]. The International Diabetes

Federation reported 4.4 and 5.2% prevalence of diabetes in Ethiopia in 2013 and 2017, respectively, among the population of 20–79 years of age [5,6]. Moreover, a systematic review revealed that the prevalence of T2DM is estimated to be 6.5% in Ethiopia [7].

Diabetes is attributed to millions fatalities annually worldwide [8]. It causes significant morbidity and suffering and lowers quality of life [9]. Inadequate glycemic control in diabetes remains a major public health issue and a risk factor for the development of diabetic complications [10,11]. Glycemic control has been demonstrated to be influenced by several factors including age, disease duration, drug use pattern and medication adherence, adherence to dietary recommendations, adherence to clinical follow-up schedules, comorbidities, and physical exercise [12–14]. Poor glycemic control in T2DM patients leads to catastrophic complications such as diabetic retinopathy, chronic kidney disease, and foot ulceration, which adversely influence the patients' HRQoL. Inability to perform physical function due to complications and feeling anxious or depressed because of high glycemic level is the cause of reduced HRQoL in diabetic patients [15]. On the other hand, close glycemic monitoring and control are vital in diabetes care as they help decrease the risk and delay the development of diabetes complications [16], and hence, improve the patients' quality of life [17]. Several factors other than glycemic control have also been shown to affect HRQoL in diabetic patients. These include demographic factors, body mass index (BMI), duration of disease, obesity, dyslipidemia, and comorbid conditions [18]. From this, it can be noted that glycemic control and HRQoL are associated with essentially overlapping factors.

Health-related quality of life shows the effect of health problems and the respective treatments on physical, emotional, and social well-being of the patients [19], and has become well recognized outcome indicator in the evaluation of the effect of diseases or the effectiveness of managements [20]. Evaluation of HRQoL is vital in that it help add to the routine clinical outcome data and guides to overall better diabetic management and improved outcomes [21,22]. Studies on HRQoL enable us to point out the perception of the patients regarding their health [23], and used as a credible indicator of unmet patient needs and the effectiveness of disease tailored managements [24]. For this end, many non-specific and disease specific tools have been developed and being used for the evaluation of HRQoL in diabetes patients [25–28]. EQ-5D tool is among the non-specific generic tools and is user-friendly and has been validated within the Ethiopian setting and disutility scores established [29,30].

Poor glycemic control is common globally, with only 50% of DM patients having their glucose controlled [7,31,32]. Currently, poor glycemic control and associated complications increase significantly in Africa [31–33]. Despite existing evidence showing glycemic control benefit diabetic patients, inadequate glycemic control is higher, especially in low and middle income countries, including in Ethiopia [34–36]. A systematic review and meta-analysis on determinants of poor glycemic control among type 2 diabetes patients in Ethiopia reported that poor glycemic control was estimated to be 61.11% [7]. These reports are indicative of glycemic control and determinant factors should be studied continually and evidence-based recommendations should be forwarded to clinicians and policy makers.

It can be depicted from the existing literatures that T2DM is a big concern globally and in Ethiopia as well. Moreover, poor glycemic control and associated complications, together with other clinical and patient-related factors, compromise the HRQoL in diabetic patients. Therefore, it is imperative to assess the glycemic control and HRQoL and determinant factors among T2DM patients. With this note, this study aimed to evaluate glycemic control and HRQoL and the determinant factors. The findings of this study may give valuable and up-to-date insights about important factors that should be considered in managing type 2 diabetes for better glycemic control and improving the HRQoL of patients receiving care.

## Methods

### The study setting and period

The study was conducted at University of Gondar Comprehensive Specialized Hospital (UoGCSH). The hospital is located in Amhara National Regional State, Northwest Ethiopia, 738 kilometers far from Addis Ababa, the capital city of the country. It serves as the sole referral, and teaching hospital in the Central Gondar Zone of the region. It serves the catchment

area with more than 13 million population in Central Gondar and neighboring zones. The hospital has 15 outpatient service departments, 28 inpatient wards, and 960 beds across different wards. The participants of the study were recruited at the chronic illness clinic of the hospital, which is one of the outpatient clinics that gives healthcare services for chronic disease patients, including diabetes. The data was collected from May 1 to July 30, 2024.

### Study design

A cross-sectional study design was employed to assess glycemic control and HRQoL and their associated factors among T2DM patients.

### Source and study population

The source population encompasses all T2DM patients who were on follow-up care at the chronic illness clinic of the hospital. The study population is the sample of subjects selected based on the sampling procedure and included in the study.

### Inclusion and exclusion criteria

Aged 18 years or older and having fasting blood glucose test results in their last three visits were eligibility criteria to include in the study, while patients who declined to provide consent for participation were immediately excluded.

### Sample size determination and sampling technique

The sample size of the study was calculated using a single population proportion formula, applying the following assumptions: proportion (p) is 62%, representing the prevalence of herbal medicine use among patients with type 2 diabetes reported in a previous study at a similar setting [37], Z score is 1.96 at 95% confidence level, and margin of error (w) is 5%. Based on these assumptions, the calculated sample size was 362 participants.

$$N = \frac{Z^2 p(1-p)}{w^2} = \frac{(1.96)^2 0.62(1-0.62)}{(0.05)^2} = 362.03 \approx 362 \text{ patients}$$

Then, 15% of the calculated sample size was added as a contingency for potential non-response rate, resulting in a final sample size of 416 patients. This study was one of the themes of a broader research project focusing on the impacts of herbal medicine use on clinical outcomes (including glycemic control) and health-related quality of life among patients with type 2 diabetes mellitus. Thus, the sample size of this specific study is based on that calculated for the project in consideration of the prevalence of herbal medicine use, which is 62% as reported in a previous study.

A daily record of the follow-up clinic showed that, on average, 20 T2DM patients attend the clinic on each working day for routine clinical follow-up and medication refill. Accordingly, it was determined that 6 patients would be selected and interviewed and their medical records reviewed each working day within 3 months of data collection period. The number of patients estimated to attend the clinic each working day was divided by the number of subjects to be selected each day. Accordingly, every third T2DM patient who attended the follow-up clinic during the data collection period was randomly approached based on the order of their attendance. Since essentially all of the patients return to the clinic within the three-month time, the three-month data collection time was preferred to give the chance of selection to perhaps all of the patients. We also carefully managed to prevent reselection of a single individual.

### Variables

#### Independent variables.

- Demographics (sex, age, residence, marital status, and educational status, occupation),

- Clinical characteristics (disease duration, adherence to dietary recommendations, physical exercise, health insurance coverage, alcohol use, comorbid conditions, diabetic complications, herbal medicine use, and antidiabetic and other medication regimens)

  **Dependent variables.**

- Glycemic control (based on the average fasting blood glucose levels over three consecutive follow-up visits)

- Health-related quality of life (EQ-5D-5L utility scores and EQ-VAS scores)

## Operational definitions

*Glycemic control:* Refers to the management of blood glucose levels within recommended targets. In this study, glycemic control was assessed using a three-visit average fasting blood glucose levels, with values $4.4 - 7.2$ mmol/L representing controlled glycemic status and values out of this range considered as uncontrolled glycemia. This definition is based on the American Diabetes Association (ADA) Professional Practice Committee Standards of Care in Diabetes—2024 [38].

*Comorbid conditions:* Coexisting diseases with type 2 diabetes mellitus such as hypertension, heart failure, rheumatoid arthritis, and other chronic illnesses.

*Diabetic complications*: Refer to medical conditions that resulted from prolonged or poorly controlled diabetes mellitus and documented in the medical records of the patients and include retinopathy, nephropathy, neuropathy, cardiovascular and cerebrovascular accidents and diabetic foot ulcer.

*Adherence to medications*: The patients' self-rating (as high, intermediate or low) of how well they follow dosage instructions to their treatment based on their personal judgment of experiences of missing doses for any reason.

*Adherence to dietary recommendation*: It includes maintaining regular mealtimes aligned with medication schedules, with a focus on controlling the intake of high glycemic index foods.

## Data collection tool, process, and quality control

A questionnaire containing socio-demographic details and clinical characteristics sections was used to collect data regarding such details. The EuroQol Five-Dimensional Five-Level scale (EQ-5D-5L) questionnaire and the EQ Visual Analogue Scale (EQ VAS), both developed by the EuroQol Group, were used to collect HRQoL data. The EQ-5D-5L tool is a generic instrument consisting of five dimensions with five levels of descriptive system questionnaires. The five dimensions are mobility, self-care, usual activities, pain/discomfort, and anxiety/depression in order. The EQ VAS scale is used to rate current health status from the patients' perspective [26]. The data collection process involved patient medical record review and face-to-face interviews. Data regarding fasting blood glucose level, comorbid conditions, and diabetes-related complications were extracted from patient medical records. Socio-demographic characteristics and clinical characteristics data (including disease duration, adherence to dietary recommendations, engagement in physical exercise, blood glucose self-monitoring, health insurance coverage, alcohol use, cigarette smoking, and herbal medicine use practice) were collected through patient interviews. To assess medication adherence, patients were asked directly to rate how well they follow their prescribed treatment, choosing from high, intermediate, or low adherence levels based on their self-judgment of missing doses for any reason. HRQoL data were also collected through face-to-face interviews using the EQ-5D-5L questionnaire and the EQ VAS.

Several quality control activities were done to ensure the reliability and validity of the data. First, the data collection instruments were well explained to the data collectors, with particular focus on variables that needed operational definitions. The data collection tool was then pretested on a small number of patients and their medical records to identify any potential ambiguities, inconsistencies, or practical problems with the tool, and some minor changes were made thereafter. To minimize selection bias, participants were randomly selected based on their appearance at the follow-up clinic,

as outlined in the sampling procedure. Lastly, each filled-out questionnaire was included in the final analysis after being checked for completeness, consistency, and eligibility criteria.

## Data analysis and interpretation

Participants were categorized based on their glycemic status (controlled or uncontrolled) as stated in the operational definitions, and factors associated with glycemic control were assessed using binary logistic regression analysis. Each dimension of the EQ-5D-5L descriptive system is rated on a five-level scale: with level 1 standing for no problems to level 5 for extreme problem or complete inability. For each participant, the EQ-5D-5L utility score was calculated using disutility coefficients, which represent the reduction in the utility score from 1 for each level of compromise in every dimension. The reduction in utility score corresponding to each level of compromise in every dimension had previously been estimated within the Ethiopian context [30]. A participant with a utility score of 1 is assumed to have no problem in any of the five dimensions, serving as the benchmark from which disutility coefficients are subtracted. The disutility coefficient increases as we go from level 2 to level 5 in each dimension. A level 1 response would not result in any reduction from perfect health state, which is represented by a utility score of 1. The equation was set up in an Excel sheet, and as the data from SPSS were transferred to the sheet the utility score was automatically calculated for each patient. To demonstrate, a participant with moderate problems in walking about, slight problems in doing his/her usual activities and with moderate pain/discomfort, and otherwise no problem in self-care and depression/anxiety dimensions (Level 1 response) would have EQ-5D-5L utility score of $1 - (0.0644715 + 0.0323013 + 0.0515949) = 0.8519206 \approx 0.85$.

The median (interquartile range, IQR) of EQ-5D-5L utility scores and EQ VAS scores were compared between subgroups because the scores were non-normally distributed across subgroups as checked using Shapiro-Wilk test ($p < 0.05$). Differences in the median scores across subgroups were determined using the Mann-Whitney U test and the Kruskal-Wallis test. Nevertheless, the overall mean and median scores were also calculated for comparison with overall results of previous similar studies. To identify factors associated with glycemic control, logistic regression analysis was carried out. Initially, a binary analysis was performed, followed by a multivariate analysis that included variables with a p-value less than 0.250 from the binary stage, as these were considered potential confounders. In addition, EQ-5D-5L utility scores were censored at 1 and the EQ VAS scores at 100, and binary Tobit regression analysis was conducted. Then multivariate Tobit regression analysis was conducted, adjusting for variables with a p-value less than 0.250 in the bivariate analysis, to examine the associated factors with HRQoL. Statistical analyses were conducted using SPSS Version 25 and STATA Version 17. Statistical differences between medians across subgroups or associations were considered significant at $p < 0.05$.

## Ethical considerations

The study protocol was reviewed by the Institutional Ethical Review Board of College of Medicine and Health Sciences, University of Gondar for ethical suitability and then approval was granted. Research, Technology Transfer, and Community Service Chief Directorate Office awarded us ethical approval letter on behalf of the review board (Ref. R/T/T/C/Eng.365/12/2023). Additionally, participants were included only after providing written informed consent to participate in the study. Furthermore, no personal identifiers were collected, and all data were handled confidentially and used solely for the purposes of the study.

## Results

A sample of 416 patients was approached; however, some were excluded because of incomplete data in their medical records, and 386 participants were included in the final analysis, resulting in a response rate of 92.8%.

## Sociodemographic characteristics

A total of 386 participants were included in the final analysis, and the proportion of males was slightly lower than females (47.9% and 52.1%, respectively). By age, 49.7% of the participants were ≤60 years old, while 50.3% of them were older

than 60 years. Out of all the respondents, about 83.4% were married and 85.6% were urban dwellers. Regarding education, the majority (36.8%) were illiterate, followed by participants with primary education (23.8%). In term of occupation, 39.6% were housewives, followed by government employees (20.5%) (Table 1).

## Clinical characteristics

The higher proportion (59.3%) of patients were with diabetic duration longer than 5 years. Metformin and glibenclamide combination was the most common antidiabetic medication regimen (35.8%), followed by metformin monotherapy (29.5%). Of the total, 37.6% participants reported that they used herbal medicine since diagnosed for diabetes. Most of the participants rated their adherence to conventional antidiabetic medications as high (65.3%), while 23.8% and 10.9% rated as intermediate and low, respectively. Almost half of the participants (47.4%) reported that they self-monitor their blood glucose levels. The majority of the respondents (67.1%) reported that they follow dietary recommendations provided by their physicians, while only 48.2% were reported engagement in physical exercise. Most participants (78.5%) had health insurance coverage. There were low reports of alcohol use (10.1%) and cigarette smoking (5.2%). Comorbid conditions were present in 68.4% of the participants, and diabetic complications were identified in 33.2% of the participants (Table 2).

## Glycemic control

**Glycemic status.** The glycemic status of each participant was determined based on the average fasting blood glucose level from three consecutive follow-up visits. The glycemic levels were categorized as per the American Diabetes Association Professional Practice Committee's Standards of Care in Diabetes−2024, which classifies glycemic control in patients with diabetes based on fasting blood glucose levels into three categories: hypoglycemia (<4.4 mmol/L), target range or controlled glycemia (4.4−7.2 mmol/L), and hyperglycemia or uncontrolled glycemia (>7.2 mmol/L). In the present study, nearly half of the respondents (51.3%) had fasting blood glucose levels of 4.4−7.2 mmol/L, which are within the

**Table 1. Socio-demographic characteristics of the participants (n = 386).**

| Variables | | Frequency | Percent |
|---|---|---|---|
| Sex | Male | 185 | 47.9 |
| | Female | 201 | 52.1 |
| Age | ≤60 years | 192 | 49.7 |
| | >60 years | 194 | 50.3 |
| Marital status | Married | 322 | 83.4 |
| | Unmarried | 64 | 16.6 |
| Residence | Urban | 329 | 85.2 |
| | Rural | 57 | 14.8 |
| Educational level | Illiterate | 142 | 36.8 |
| | Primary education | 92 | 23.8 |
| | Secondary education | 68 | 17.6 |
| | Tertiary education | 84 | 21.8 |
| Occupation | Farmer | 35 | 9.1 |
| | House wife | 153 | 39.6 |
| | Government employee | 79 | 20.5 |
| | Private employee | 48 | 12.4 |
| | Trader | 44 | 11.4 |
| | Other | 27 | 7.0 |

**Table 2. Clinical characteristics of the participants (n = 386).**

| Variables | | Frequency | Percent |
|---|---|---|---|
| Duration since diagnosed for diabetes | ≤5 years | 157 | 40.7 |
| | >5 years | 229 | 59.3 |
| Diabetes medication regimen | Metformin | 114 | 29.5 |
| | Metformin and glibenclamide | 138 | 35.8 |
| | Metformin and NPH insulin | 59 | 15.3 |
| | NPH insulin | 62 | 16.1 |
| | Others | 13 | 3.4 |
| Herbal medicine use | Yes | 145 | 37.6 |
| | No | 241 | 62.4 |
| Adherence to conventional medications | High | 252 | 65.3 |
| | Intermediate | 92 | 23.8 |
| | Low | 42 | 10.9 |
| Self-blood glucose monitoring | Yes | 183 | 47.4 |
| | No | 203 | 52.6 |
| Follow dietary recommendations | Yes | 259 | 67.1 |
| | No | 127 | 32.9 |
| Physical exercise | Yes | 186 | 48.2 |
| | No | 200 | 51.8 |
| Health insurance coverage | Yes | 303 | 78.5 |
| | No | 83 | 21.5 |
| Alcohol use | Yes | 39 | 10.1 |
| | No | 347 | 89.9 |
| Cigarette smoking | Yes | 20 | 5.2 |
| | No | 366 | 94.8 |
| Comorbid condition | Yes | 264 | 68.4 |
| | No | 122 | 31.6 |
| Diabetic complication | Yes | 128 | 33.2 |
| | No | 258 | 66.8 |

recommended range. However, a considerable proportion (47.9%) of patients were with glycemic levels exceeding 7.2 mmol/L, indicating poor glycemic control. Blood glucose levels below 4.4 mmol/L were detected in only 0.8% of participants, indicating the possibility of hypoglycemia (Fig 1).

**Factors associated with glycemic control.** In the multivariate logistic regression analysis, a number of variables were found to have a significant association with glycemic control. Older age was associated with poor glycemic control, with patients older than 60 years were 2.802 times more likely (AOR: 2.802, 95% CI: (1.631−4.815), p < 0.001) to have uncontrolled glycemia compared to those with aged 60 years or younger. Similarly, patients who were longer than five years since they diagnosed with T2DM were 2.647 times more likely (AOR: 2.647, 95% CI: (1.503−4.663), p < 0.01) to have uncontrolled blood glucose level compared to those who were with shorter time since diagnosed for the disease. Adherence to dietary recommendations was predictive of good glycemic management. Individuals who reported adherence to dietary recommendations had 49.0% lower odds of having uncontrolled blood glucose levels (AOR: 0.510, 95% CI: (0.285–0.910), p < 0.05). Patients with comorbid conditions were 2.133 times more likely to have uncontrolled blood glucose level (AOR: 2.133, 95% CI: (1.213−3.754), p < 0.01) compared to those without comorbid conditions. Likewise, patients with diabetes complications were 2.816 times more likely (AOR: 2.816, 95% CI: (1.124−4.508),

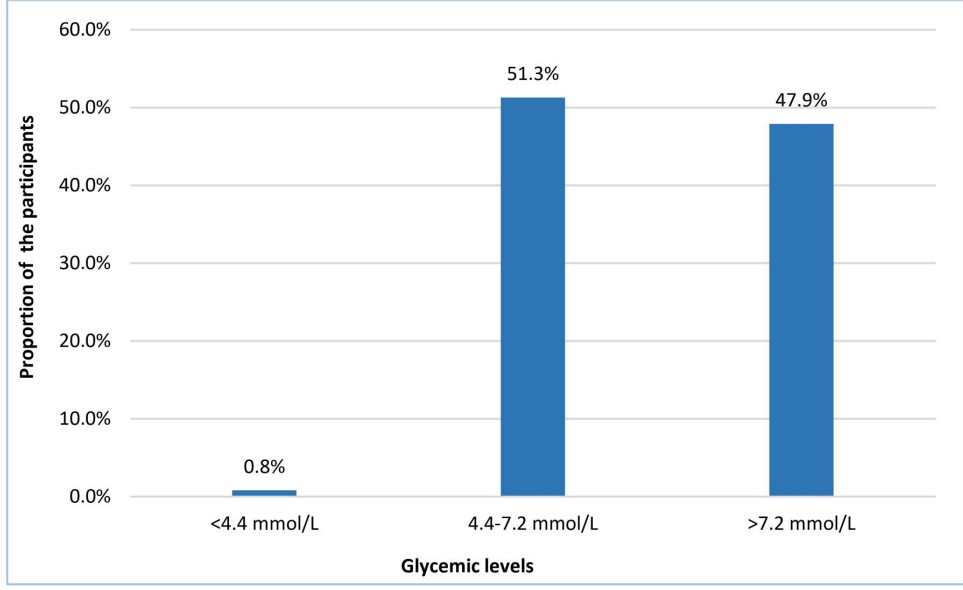

**Fig 1. Percentage distribution of the T2DM patients by glycemic levels (n = 386).**

p < 0.05) to have uncontrolled blood sugar levels compared to those without complications. Furthermore, high adherence to antidiabetic medications was strongly associated with better glycemic control; the odds of having uncontrolled glycemia was reduced by more than 75.0% in patients rated their level of adherence as high compared to in those rated as low (AOR: 0.223, 95% CI: (0.091 − 0.545), p < 0.01). Patients who reported using herbal medicines were 3.074 times more likely (AOR: 3.074, 95% CI: (1.786 − 5.292), p < 0.001) to have uncontrolled blood glucose level compared to those who didn't use herbal medicines (Table 3).

**Health-related quality of life**

**Proportions of responses by level of severity for EQ-5D-5L dimensions.** Responses to EQ-5D-5L dimensions showed that a considerable proportion of the participants had perceived problems across the different dimensions. In the mobility dimension, 46.4% of the participants experienced some level mobility problem, essentially slight or moderate problem in walking about. Problems in self-care were relatively uncommon; only 30.1% of patients reported having difficulty in self-care, and all reported problems were slight (most) or moderate level. However, 76.2% of the patients reported that they experience problems in performing their usual activities. The level of difficulties in this regard ranges from slight problem (mostly) to complete inability. Pain/discomfort was the most affected domain, with 85.8% of respondents reporting having problems in the domain. The degree of the problems in this domain were essentially slight (52.3%) followed by moderate (24.6%). Similarly, problems related to anxiety/depression affected 74.6% of the patients. Only 7.3% individuals reported perfect health status (Fig 2).

**Distribution of EQ-5D-5L utility and EQ VAS scores by glycemic status.** EQ-5D-5L utility and EQ VAS scores distributions differed considerably between cases with controlled and uncontrolled blood glucose levels. Individuals with controlled glycemia more frequently reported higher EQ-5D-5L utility and EQ VAS scores, showing they had better HRQoL. Relatively greater number of cases were concentrated around the higher ends of the EQ-5D-5L utility and EQ VAS scores. Conversely, the distribution of scores for participants with uncontrolled blood glucose levels were broader and relatively left-skewed, with a higher number of cases in the group reported lower scores. These variations imply that

**Table 3. Logistic regression analysis of factors associated with glycemic control (n = 386).**

| Factors | | Glycemic status | | AOR (95% CI) | p value |
|---|---|---|---|---|---|
| | | Controlled | Uncontrolled | | |
| Age (year) | ≤60 | 133 | 59 | 1 | 0.000 |
| | >60 | 65 | 129 | 2.802 (1.631 − 4.815)*** | |
| Disease duration (year) | ≤5 | 111 | 46 | 1 | 0.001 |
| | >5 | 87 | 142 | 2.647 (1.503 − 4.663)** | |
| Sex | Male | 97 | 88 | 0.557 (0.252-1.232) | 0.149 |
| | Female | 101 | 100 | 1 | |
| Residence | Urban | 167 | 162 | 1.250 (0.490 − 3.185) | 0.641 |
| | Rural | 31 | 26 | 1 | |
| Marital status | Married | 166 | 156 | 0.687 (0.334 − 1.415) | 0.309 |
| | Unmarried | 32 | 32 | 1 | |
| Health insurance coverage | Yes | 151 | 152 | 1.289 (0.246 − 2.332) | 0.478 |
| | No | 47 | 36 | 1 | |
| Follow dietary recommendations | Yes | 151 | 108 | 0.510 (285 − 0.910)* | 0.023 |
| | No | 47 | 80 | 1 | |
| Alcohol use | Yes | 23 | 16 | 0.704 (0.275 − 1.804) | 0.464 |
| | No | 175 | 172 | 1 | |
| Physical exercise | Yes | 120 | 66 | 0.622 (0.367 − 1.053) | 0.077 |
| | No | 78 | 122 | 1 | |
| Comorbidity | Yes | 115 | 149 | 2.133 (1.213 − 3.754)** | 0.009 |
| | No | 83 | 39 | 1 | |
| Diabetic complication | Yes | 40 | 88 | 2.816 (1.124 − 4.508)* | 0.039 |
| | No | 158 | 100 | 1 | |
| Blood glucose self-monitoring | Yes | 102 | 81 | 0.809 (0.481 − 1.360) | 0.424 |
| | No | 96 | 107 | 1 | |
| Adherence to antidiabetic medications | High | 150 | 102 | 0.223 (0.091 − 0.545)** | 0.001 |
| | Intermediate | 35 | 57 | 0.495 (0.185 − 1.325) | |
| | Low | 13 | 29 | 1 | |
| Herbal medicine use | Yes | 44 | 101 | 3.074 (1.786 − 5.292)*** | 0.000 |
| | No | 154 | 87 | 1 | |

* p < 0.05, ** p < 0.01, *** p < 0.001.

perceived health-related quality of life was notably lower in patients with uncontrolled glycemia compared with in patients with controlled glycemic levels (Fig 3).

**Subgroup comparison of EQ-5D-5L utility and EQ-VAS scores.** The overall median (IQR) EQ-5D-5L utility score was 0.86 (0.76–0.93), while the EQ VAS score was 75.0 (65.0–80.0). The non-parametric Mann-Whitney U and Kruskal-Wallis tests revealed significant differences in median EQ-5D-5L utility and EQ-VAS scores across various patient subgroups. Patients aged ≤60 years scored higher median EQ-5D-5L utility score (p < 0.001) and EQ VAS score (p < 0.001) compared to those with older age. The median EQ-5D-5L utility (p < 0.01) and EQ VAS (p < 0.001) scores of patients with diabetes duration of 5 years or shorter were significantly higher than those of patients with longer disease duration. Patients reported to engage in physical activity had significantly higher median EQ-5D-5L utility (p < 0.001) and EQ VAS (p < 0.05) scores. Respondents who were adherent to dietary recommendations also had significantly higher median EQ-5D-5L utility (p < 0.01) and EQ VAS (p < 0.05) scores. Patients who self-monitor their own blood sugar levels

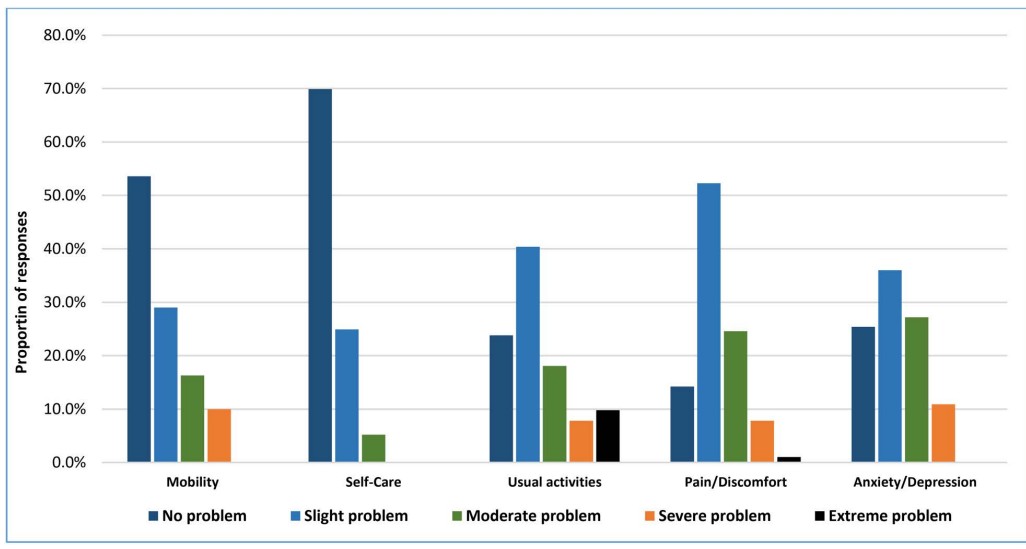

**Fig 2. Proportions of responses by level of severity for EQ-5D-5L dimensions (n = 386).**

showed significantly higher median EQ-5D-5L utility score (p < 0.05), whereas no significant difference was observed in the median EQ VAS score. Participants with controlled glycemia had significantly higher median EQ-5D-5L utility and EQ VAS scores (p < 0.001). In contrast, the median EQ-5D-5L utility and EQ VAS scores were significantly lower (p < 0.01) in participants with comorbid conditions. Similarly, the median EQ-5D-5L utility (p < 0.01) and EQ VAS score (p < 0.001) were significantly reduced among respondents with diabetic complications. In addition, those reported using herbal remedies had significantly lower (p < 0.001) median EQ-5D-5L utility and EQ VAS scores compared to non-users. Differences at least between the highest and the lowest median EQ-5D-5L utility scores of subgroups of antidiabetic medication regimens (p < 0.05) and adherence to antidiabetic medications (p < 0.01) were also observed. Likewise, there were significant differences in the median EQ VAS scores among subgroups of antidiabetic medication regimens (p < 0.05) and adherence to antidiabetic medications (p < 0.001) (Table 4).

**Factors associated with HRQoL.** Multivariate Tobit regression analysis showed that many factors were significantly associated with HRQoL as reflected by correlations with EQ-5D-5L utility and EQ VAS scores. Older age (β = −0.131, 95% CI: (−0.172,-0.090), p < 0.001), diabetes duration (β = −0.069, 95% CI: (−0.112,-0.027), p < 0.001), comorbid conditions (β = −0.042, 95% CI: (−0.073, −0.011), p < 0.05), and diabetic complications (β = −0.061, 95% CI: (−0.109,-0.014), p < 0.05) were significantly negatively associated with EQ-5D-5L utility scores. Herbal medicine use was also significantly negatively correlated (β = −0.165, 95% CI: (−0.205,-0.124), p < 0.001) with EQ-5D-5L utility scores. On the other hand, physical activity (β = 0.105, 95% CI: (0.024,0.144), p < 0.001), and glycemic control (β = 0.103, 95% CI: (0.032, 0.163), p < 0.01) were shown to be positively associated with EQ-5D-5L utility scores. Combination therapies with metformin and glibenclamide (β = 0.075, 95% CI: (0.029, 0.122), p < 0.01) and metformin and NPH insulin (β = 0.090, 95% CI: (0.032,0.149), p < 0.01) were also positively associated with EQ-5D-5L scores compared to with metformin alone. Adherence was also positively correlated with EQ-5D-5L scores. Low (β = −0.086, 95% CI: (−0.148,-0.024), p < 0.01) and intermediate (β = −0.046, 95% CI: (−0.090,-0.001), p < 0.05) adherence ratings to conventional antidiabetic medications were significantly associated with lower EQ-5D-5L utility scores compared to high adherence. Many of the factors associated with EQ-5D-5L utility scores were also associate with EQ VAS scores. Older age (β = −6.093, 95% CI: (−8.465,-3.721), p < 0.001), comorbid illnesses (β = −2.764, 95% CI: (−4.972,-0.526), p < 0.05), and diabetic complications (β = −2.507, 95% CI: (−4.645,-0.369), p < 0.05) were significantly negatively correlated with EQ VAS scores.

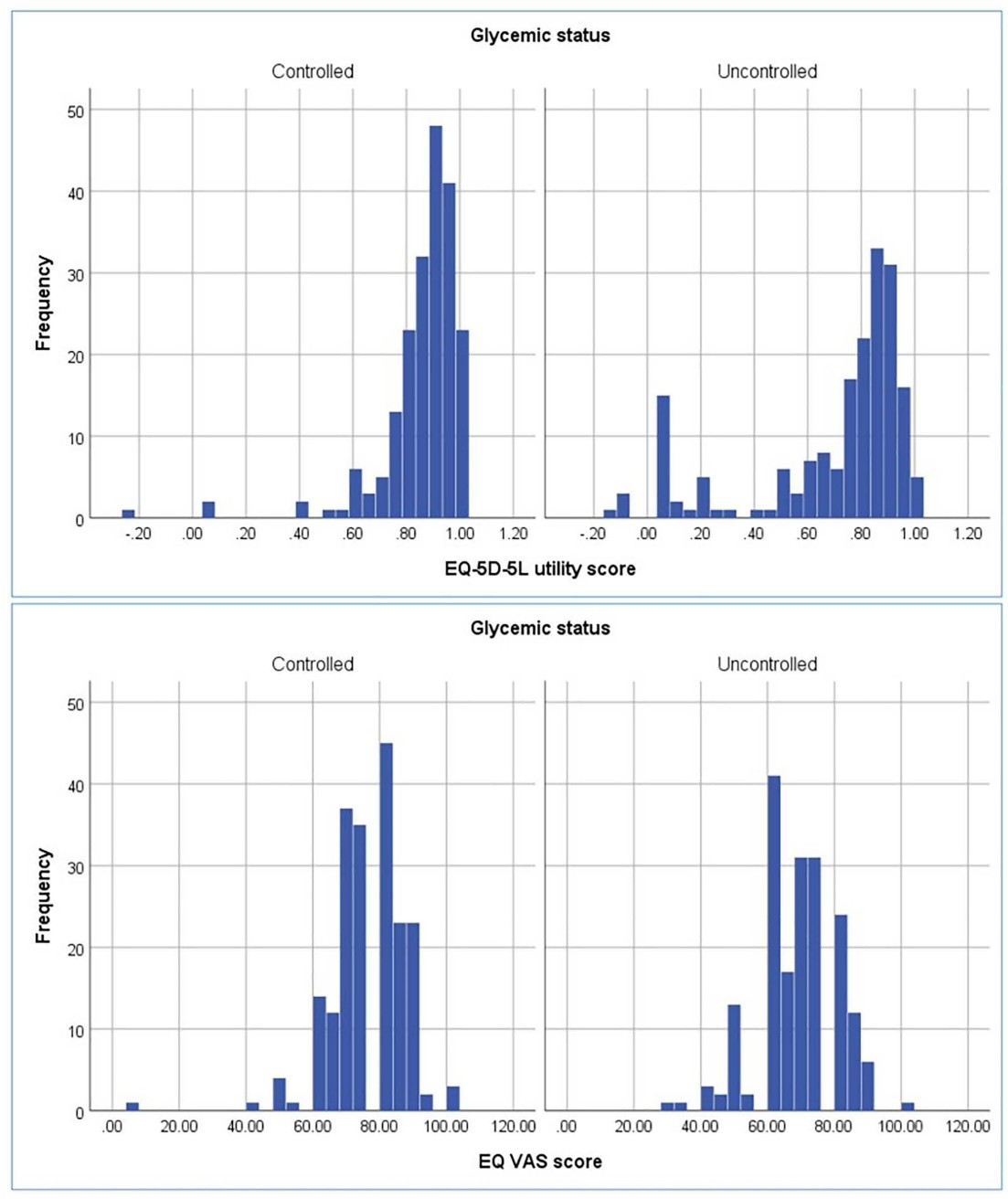

**Fig 3. Frequency distribution of EQ-5D-5L utility and EQ VAS scores by glycemic status (n = 386).**

Likewise, herbal medicine use was negatively correlated with EQ VAS score. Patients who reported herbal medicine use scored 2.754 points lower on EQ VAS scale compared to non-users (β = −2.754, 95% CI: (−5.097,-0.412), p < 0.05). Conversely, engagement in physical exercise (β = 6.019, 95% CI: (2.118,9.920), p < 0.01) and having controlled blood glucose levels (β = 2.856, 95% CI: (0.307,5.404), p < 0.05) were shown to significantly positively correlated with EQ VAS scores. Adherence was also positively associated with EQ VAS score. Compared to high adherence, low (β = −6.724, 95%

**Table 4. Median differences of EQ-5D-5L utility and EQ-VAS scores with patient profiles (result from Independent-Samples Mann-Whitney U and Kruskal-Wallis Test) (n = 386).**

| Variables | | EQ-5L-5D utility score | | | EQ-VAS score | | |
|---|---|---|---|---|---|---|---|
| | | Median (IQR) | Mean rank | p value | Median (IQR) | Mean rank | p value |
| Age | ≤60 years | 0.91 (0.85-0.96) | 248.00 | **0.000** | 75.0 (70.0-85.0) | 236.58 | 0.000 |
| | >60 years | 0.82 (0.63-0.88) | 139.56 | | 70.0 (60.0-75.0) | 150.87 | |
| Sex | Male | 0.86 (0.75-0.91) | 187.39 | 0.344 | 70.0 (65.0-80.0) | 176.80 | 0.156 |
| | Female | 0.87 (0.77-0.93) | 198.21 | | 70.0 (65.0-85.0) | 182.37 | |
| Residence | Urban | 0.87 (0.76-0.93) | 195.22 | 0.467 | 75.0 (65.0-80.0) | 195.59 | 0.372 |
| | Rural | 0.85 (0.74-0.91) | 183.58 | | 75.0 (62.5-80.0) | 181.44 | |
| Marital status | Married | 0.85 (0.76-0.93) | 189.35 | 0.101 | 75.0 (65.0-80.0) | 186.94 | 0.067 |
| | Unmarried | 0.89 (0.81-0.94) | 214.40 | | 75.0 (70.0-85.0) | 206.48 | |
| Educational level | Illiterate | 0.85 (0.70-0.93) | 177.33 | 0.102 | 75.0 (63.8-80.0) | 187.67 | 0.584 |
| | Primary education | 0.88 (0.79-0.94) | 211.86 | | 75.0 (65.0-80.0) | 187.41 | |
| | Secondary education | 0.87 (0.76-0.91) | 190.21 | | 75.0 (65.0-80.0) | 207.39 | |
| | Tertiary education | 0.87 (0.80-0.91) | 203.40 | | 70.0 (65.0-84.3) | 198.79 | |
| Occupation | Farmer | 0.85 (0.74-0.91) | 176.83 | 0.339 | 75.0 (65.0-80.0) | 187.63 | 0.635 |
| | House wife | 0.85 (0.77-0.94) | 197.42 | | 75.0 (65.0-80.0) | 201.95 | |
| | Government employee | 0.88 (0.83-0.93) | 211.32 | | 70.0 (65.0-80.0) | 196.52 | |
| | Private employee | 0.85 (0.77-0.93) | 190.61 | | 75.0 (65.0-80.0) | 195.48 | |
| | Trader | 0.82 (0.64-0.93) | 166.24 | | 70.0 (60.0-80.0) | 172.81 | |
| | Other | 0.86 (0.77-0.91) | 190.30 | | 72.5 (60.0-80.0) | 174.57 | |
| Health insurance coverage | Yes | 0.86 (0.76-0.93) | 191.61 | 0.525 | 75.0 (65.0-80.0) | 191.62 | 0.523 |
| | No | 0.88 (0.76-0.94) | 200.39 | | 75.0 (70.0-80.0) | 200.36 | |
| Follow dietary recommendations | Yes | 0.87 (0.78-0.94) | 205.15 | 0.003 | 75.0 (65.0-80.0) | 201.56 | 0.041 |
| | No | 0.85 (0.71-0.91) | 169.75 | | 70.0 (60.0-80.0) | 177.06 | |
| Alcohol use | Yes | 0.85 (0.80-0.92) | 192.00 | 0.929 | 75.0 (65.0-80.0) | 182.67 | 0.518 |
| | No | 0.87 (0.76-0.93) | 193.67 | | 75.0 (65.0-80.0) | 194.72 | |
| Physical exercise | Yes | 0.90 (0.83-0.94) | 231.06 | 0.000 | 75.0 (70.0-80.0) | 205.52 | 0.039 |
| | No | 0.83 (0.64-0.90) | 158.57 | | 70.0 (60.0-80.0) | 182.32 | |
| Diabetes duration since diagnosis | ≤5 years | 0.90 (0.84-0.94) | 233.76 | 0.005 | 75.0 (70.0-83.5) | 227.84 | 0.000 |
| | >5 years | 0.83 (0.67-0.91) | 165.90 | | 70.0 (60.0-80.0) | 169.95 | |
| Comorbid conditions | Yes | 0.82 (0.68-0.90) | 176.57 | 0.001 | 70.0 (59.3-75.0) | 177.25 | 0.003 |
| | No | 0.88 (0.81-0.93) | 208.50 | | 75.0 (70.0-85.0) | 212.30 | |
| Diabetes complications | Yes | 0.83 (0.66-0.90) | 161.61 | 0.001 | 70.0 (60.0-75.0) | 157.41 | 0.000 |
| | No | 0.88 (0.79-0.93) | 209.32 | | 75.0 (70.0-80.0) | 211.41 | |
| Blood glucose self-monitoring | Yes | 0.88 (0.79-0.94) | 207.00 | 0.024 | 75.0 (70.0-80.0) | 204.22 | 0.165 |
| | No | 0.85 (0.75-0.91) | 181.33 | | 75.0 (65.0-80.0) | 190.15 | |
| Antidiabetic medications | Metformin | 0.83 (0.67-0.91) | 168.96 | 0.030 | 75.0 (65.0-80.0) | 187.70 | 0.038 |
| | Metformin and glibenclamide | 0.87 (0.80-0.94) | 208.47 | | 75.0 (65.0-85.0) | 208.55 | |
| | Metformin and NPH insulin | 0.88 (0.82-0.93) | 209.25 | | 75.0 (70.0-80.0) | 206.99 | |
| | NPH insulin | 0.85 (0.72-0.96) | 197.99 | | 70.0 (65.0-80.0) | 169.86 | |
| | Others | 0.85 (0.65-0.90) | 156.88 | | 70.0 (55.0-77.0) | 136.04 | |
| Glycemic status | Controlled | 0.89 (0.81-0.94) | 222.35 | 0.000 | 75.0 (70.0-75.0) | 226.86 | 0.000 |
| | Uncontrolled | 0.83 (0.67-0.90) | 162.48 | | 70.0 (60.0-70.0) | 157.63 | |
| Adherence to antidiabetic medications | High | 0.89 (0.79-0.94) | 212.05 | 0.001 | 75.0 (70.0-80.0) | 211.28 | 0.000 |
| | Moderate | 0.83 (0.76-0.89) | 158.08 | | 70.0 (60.0-75.0) | 157.11 | |
| | Low | 0.83 (0.59-0.91) | 159.82 | | 70.0 (60.0-80.0) | 166.51 | |

*(Continued)*

**Table 4.** (Continued)

| Variables | | EQ-5L-5D utility score | | | EQ-VAS score | | |
|---|---|---|---|---|---|---|---|
| | | Median (IQR) | Mean rank | p value | Median (IQR) | Mean rank | p value |
| Herbal medicine use | Yes | 0.77 (0.50-0.87) | 125.65 | 0.000 | 70.0 (60.0-75.0) | 158.68 | 0.000 |
| | No | 0.90 (0.83-0.94) | 234.32 | | 75.0 (70.0-80.0) | 214.45 | |

IQR, interquartile range.

CI: (−10.270,-3.178), p < 0.001) and intermediate (β = −3.679, 95% CI: (−6.226,-1.133), p < 0.01) adherence ratings to conventional antidiabetic medications were significantly associated with lower VAS scores (Table 5).

## Discussion

Poor glycemic control in diabetes remains a notable public health problem and is associated with various determinant factors [11–15]. Inadequate blood glucose control in diabetic patients often leads to serious complications that may negatively impact HRQoL [15]. HRQoL assessments help determine how diseases and associated complications and treatments affect overall health status [39,40]. Glycemic control and several other factors influence HRQoL

**Table 5. Tobit regression analysis of factors associated with EQ-5L-5D utility and VAS scores (n = 386).**

| Factors (reference) | | EQ-5L-5D utility score | | EQ VAS score | |
|---|---|---|---|---|---|
| | | β-coefficient (95% CI) | p value | β-coefficient (95% CI) | p value |
| Age (≤ 60 years) | >60 years | −0.131 (−0.172,-0.090)*** | 0.000 | −6.093 (−8.465,-3.721)*** | 0.000 |
| Sex (Female) | Male | −0.010 (−0.066,0.046) | 0.718 | −2.247 (−5.464,0.970) | 0.170 |
| Residence (Rural) | Urban | 0.035 (−0,027,0.096) | 0.264 | 2.904 (−0.639,6.447) | 0.108 |
| Marital status (Unmarried) | Married | −0.004 (−0.055,0.046) | 0.867 | −2.683 (−5.598,0.232) | 0.071 |
| Occupation (Farmer) | House wife | −0.017 (−0.108,0.074) | 0.717 | −2.986 (−8.247,2.276) | 0.265 |
| | Government employee | 0.023 (−0.063,0.109) | 0.600 | −1.328 (−6.282,3.628) | 0.598 |
| | Private employee | −0.040 (−0.132,0.052) | 0.391 | −0.508 (−6.055,4.793) | 0.851 |
| | Trader | −0.033 (−0.122,0.057) | 0.472 | −1.646 (−6.790,3.497) | 0.529 |
| | Other | 0.000 (−0.102,0.102) | 0.996 | −2.241 (−8.127,3.646) | 0.455 |
| Follow dietary recommendations (No) | Yes | 0.013 (−0.028,0.055) | 0.527 | 2.007 (−0.024,4.038) | 0.424 |
| Physical exercise (No) | Yes | 0.105 (0.024,0.144)*** | 0.000 | 6.019 (2.118,9.920)** | 0.002 |
| Diabetes duration since diagnosis (≤5 years) | >5 years | −0.069 (−0.112,-0.027)** | 0.001 | −2.374 (−4.803,0.056) | 0.055 |
| Comorbid conditions (No) | Yes | −0.042 (−0.073, −0.011)* | 0.034 | −2.764 (−4.972,-0.526)* | 0.043 |
| Diabetes complication (No) | Yes | −0.061 (−0.109, −0.014)* | 0.012 | −2.507 (−4.645,-0.369)* | 0.049 |
| Self-monitoring of blood sure level (No) | Yes | 0.013 (−0.024,-0.050) | 0.496 | 0.080 (−0.187,0.179) | 0.061 |
| Antidiabetic medications (Metformin) | Metformin and glibenclamide | 0.075 (0.029,0.122)** | 0.002 | 1.927 (−0.742,4.596) | 0.157 |
| | Metformin and NPH insulin | 0.090 (0.032,0.149)** | 0.003 | 2.117 (−1.260,5.493) | 0.218 |
| | NPH insulin | 0.025 (−0.033,0.083) | 0.396 | −2.346 (−5.688,0.996) | 0.168 |
| | Other | −0.029 (−0.077,0.136) | 0.590 | −4.063 (−10.203,2.077) | 0.194 |
| Glycemic status (Uncontrolled) | Controlled | 0.103 (0.032,0.163)** | 0.009 | 2.856 (0.307,5.404)* | 0.028 |
| Adherence to antidiabetic medications (High) | Intermediate | −0.046 (−0.090,-0.001)* | 0.043 | −3.679 (−6.226,-1.133)** | 0.005 |
| | Low | −0.086 (−0.148,-0.024)** | 0.006 | −6.724 (−10.270,-3.178)*** | 0.000 |
| Herbal drug use (No) | Yes | −0.165 (−0.205,-0.124)*** | 0.000 | −2.754 (−5.097,-0.412)* | 0.021 |

* p < 0.05, ** p < 0.01, *** p < 0.001.

in diabetic patients [15,18]. This study aimed to assess glycemic control and HRQoL in T2DM patients and outlined determinant factors. The results were compared with those of previous similar studies for better understanding of the findings.

Based on the American Diabetic Association classification, the prevalence of uncontrolled glycemia was found to be 48.7%. This result is comparable to that of a previous study in Ethiopia, which reported 45.2% poor glycemic control [41], and is higher than another study in the country, which reported 41.6% prevalence of poor glycemic control [42]. On the other hand, it is considerably lower than the results reported by many other studies in Ethiopia, which are 72.8%, 66.2%, 58.1%, and 72.6% [33,43–45]. It is also noticeably lower compared to findings from other nations, including Kenya, 81.6% [46], Tanzania, 66.1% [13], Malaysia, 59.2% [47], and Saudi Arabia, 74.9% [48]. Differences in sociodemographic and clinical characteristics of the patients and healthcare services, together with environmental and lifestyle factors, may account for the discrepancies in the prevalence of poor glycemic control between different studies [41]. Besides, the differences between the results in this and in other studies could be due to differences in study design and sample size, and parameters used for categorization of glycemic control status (FBG levels vs. HbA1c) [42].

As presented in Table 3, multivariate logistic regression demonstrated that several factors were associated with uncontrolled blood glucose. Patients older than 60 were 2.802 times more likely (p < 0.01) to have uncontrolled blood glucose than younger individuals. This result is consistent with a previous finding [44]. Similarly, patients who were longer than 5 years since they diagnosed with T2DM were 2.647 times more likely (p < 0.01) to have uncontrolled blood glucose level. This is in line with prior studies [7,33,48,49], and the possible explanation is that pancreatic β-cells may gradually be depleted and play a significant role in poor glycemic control among T2DM patients with longer duration of the disease [50]. Comorbid conditions and diabetic complications were associated with 2.133 and 2.816 times likelihood of having uncontrolled blood glucose levels, respectively (p < 0.05). Both or either of these associations were shown by a number of previous studies [7,44,47,50–52], possibly due to complex treatment regimens that negatively affect adherence.

Adherence to dietary recommendations was associated with a 49.0% decrease in the odds of uncontrolled blood glucose (p < 0.05). This result is in agreement with those of other studies [42,53]. Moreover, this finding supports existing evidence indicating that dietary modification is one of the approaches for regulating blood glucose levels [54]. The likelihood of having uncontrolled glycemia was decreased by more than 75.0% in patients who rated their level of adherence to antidiabetic medications as high compared to those rated as low (p < 0.01). Previous studies have reported that poor adherence to antidiabetic medications is associated with poor glycemic control [7,51,53,55], and hence, as demonstrated in this study, good adherence to antidiabetic medications significantly improves glycemic control. The odds of having uncontrolled blood sugar level was 3.074 in patients who reported using herbal medicines (p < 0.001). This may be because those with uncontrolled glycemic levels and related complications are more likely to seek complementary and alternative medicines to supplement their treatments. A previous study reported that herbal medicine use practice was more likely among T2DM patients with diabetic complications [56]. Another study also reported that patients who used herbal medicine had significantly higher HbA1c levels relative to non-users [57].

In HRQoL assessment, pain/discomfort was the most affected of all EQ-5D-5L dimensions, in which 85.8% of respondents reported from slight to extreme problems. This result is in agreement with similar studies in Ethiopia, China, Nigeria, and Iran [58–64], whereas in discrepancy with another study conducted in India, reporting anxiety/depression was the most affected dimension, indicating 91.0% reported slight to severe problem [65]. This shows that pain/discomfort dimension should get due attention and be managed properly in addition to regular treatments for diabetes and other comorbidities. Doing usual activities was the second most affected EQ-5D-5L dimension, with 76.2% of the participants reported slight problem to inability to walk about. Likewise, this dimension was reported as the second or third most common problem in similar studies [59,61], implicating the burden of diabetes in impeding performing daily livelihood activities. Anxiety/

depression is the third most affected dimension, in which 74.6% of the patients reported some level of problem. This proportion is less than the report from India (91.0%) [65], whereas, by far, greater than the results of other earlier studies in China (23.5%), Nigeria (57.6%), Iran (56.6%), and Japan (25.7%) [60,61,63,66]. This shows that anxiety/depression dimension is also highly affected and needs due consideration and assessment during routine follow-up visits, and eligible patients should be linked to psychiatric care services.

The overall mean EQ-5D-5L utility score was found to be 0.78, and is similar to the report of a similar study in Indonesia (0.77) [67], whereas lower than the results of other similar studies in Ethiopia (0.87) and other nations including Japan (0.86 and 0.901) [66,68], Korea (0.87) [69], Finland (0.85) [70], and in Birjand, Iran (0.89) [63]. The mean EQ VAS score was calculated to be 72.4, while other similar studies reported 56.8 [64], 65.22 [63], and 76.34 [58] scores in this regard. The overall median EQ-5D-5L utility and EQ VAS scores were 0.86 and 75.0, respectively, and are lower than reported by a study in Ethiopia, 0.95 and 80.0 [58]. On the other hand, the overall median EQ-5D-5L utility score of this study is higher than that reported by a study in Nigeria (0.77), whereas the median EQ VAS score is lower than reported by the same study (80.0) [61]. The variations in the scores between studies may stem from differences in socioeconomic and clinical characteristics of the participants, health services, and the disutility value sets used. Moreover, the mean EQ-5D-5L utility and VAS scores are less than the results for the general population in Ethiopia, reported to be 0.94 and 87.26, respectively [30]. This difference is in line with the existing literatures showing that diabetes impairs HRQoL due to hyperglycemia and associated complications [9–11].

In line with prior studies [58,61,63,64,71] we found statistically significant differences in median EQ-5D-5L utility and EQ VAS scores across different patient subgroups including between groupings in age, adherence to dietary recommendations, duration of disease, comorbidities, diabetic complications, self-monitoring of blood glucose level, antidiabetic medication regimens, glycemic status (controlled vs. uncontrolled), and adherence to antidiabetic medications. The results also demonstrated that the median EQ-5D-5L utility and EQ VAS scores of herbal medicine users and non-users were significantly different, with the scores of herbal medicine users being significantly smaller. This could be because patients with poor glycemic control and negatively affected HRQoL were more likely to seek supplemental or alternative interventions. A previous study showed that the likelihood of herbal medicine use was significantly higher among diabetic patients having diabetic complications [56], and, in turn, another study reported that herbal medicine users were more likely to have poorly controlled glycemic levels [57].

Multivariate Tobit regression model (Table 5) demonstrated that glycemic control, physical activity, and high adherence to antidiabetic medications were significantly positively associated with EQ-5D-5L utility and EQ VAS scores. These positive associations are in agreement with those reported by other similar studies [72–74]. Inadequate glycemic control in T2DM patients leads to complications that may adversely affect their HRQoL [15]. In line with this, our finding showed that controlled glycemia was associated with improved HRQoL scores. Physical activity activates brain chemicals associated with improved mood and enhances self-efficacy and self-esteem, and helps alleviate depressive symptoms [75]. This may help counteract the negative impacts of diabetes and other diseases on HRQoL. Strict adherence to antidiabetic medications could result in better glycemic control and decrease the progression of diabetic complications, ultimately improving HRQoL. Older age, diabetes duration, comorbidities, and diabetic complications were significantly associated with lower HRQoL scores. Previous studies also demonstrated that these variables are negatively associated with HRQoL scores [58,59,62,63,68,71]. Mental and physical limitations and the chance of having comorbidities and other debilitating conditions increase with age [61,76]. These conditions, in concert with the diabetes, may considerably negatively affect the patients' HRQoL. As outlined by the American Diabetes Association [38], longer duration of diabetes is associated with higher probabilities of developing various complications, which in turn may result in reductions in HRQoL scores. Interestingly, herbal medicine use practice was also significantly correlated with lower EQ-5D-5L utility and EQ VAS scores. This result is in line with previous studies that reported patients who use herbal medicines are more likely to have diabetes complications and poor glycemic control [56,57].

## Strengths and limitations

The study has strengths and limitations. Assessing glycemic control and HRQoL simultaneously could promote a more holistic understanding of determinant factors affecting both outcomes. Besides, the study assessed the influence of glycemic control on HRQoL, showing the importance of glycemic control for the general well-being of diabetic patients. Utilizing the average of FBG test results from three consecutive follow-up visits may improve the reliability of the glycemic data and, consequently, the conclusions drawn about glycemic control. This study has some limitations that should be accounted in interpreting the findings. First, as it is a cross-sectional study, it does not show the nature of interaction between the influencing factors and the outcome variables. Second, the results may not be generalizable to other patients with type 2 diabetes mellitus in Ethiopia or other regions because it was conducted at a single setting.

## Conclusion

In conclusion, nearly half of the patients were with uncontrolled glycemia and the overall HRQoL score were lower than that of the general population in Ethiopia. Several factors were correlated with both glycemic control and HRQoL. Adherence to antidiabetic medication was positively associated with both glycemic control and HRQoL. In contrast, older age, longer duration of diabetes, presence of comorbidities, diabetic complications, and use of herbal medicine were all negatively associated with both outcomes. On the other hand, adherence to dietary recommendations was positively associated only with glycemic control, while engagement in physical exercise was positively associated only with HRQoL. Furthermore, glycemic control was associated with improved HRQoL. The findings underscore the importance of interventions targeting modifiable factors, such as dietary modifications, physical activity, and adherence support, to improve overall glycemic control and HRQoL. Moreover, most patients reported problems in pain/discomfort, usual activity, and anxiety/depression dimensions of HRQoL. These findings dictate the need to give due attention to these dimensions in patient management.

## Supporting information

**S1 File. Raw data.**
(XLSX)

## Acknowledgments

We extend our sincere gratitude to the College of Medicine and Health Sciences Community Service, Research and Technology Transfer Directorate and the clinical director office of the hospital for facilitating the official procedures to obtain authorization for conducting the study at the hospital. We are also grateful to all healthcare providers who were working at the chronic illness clinic of the hospital during the data collection for their sincere collaboration in the data collection process.

## Author contributions

**Conceptualization:** Assefa Belay Asrie, Tafere Mulaw Belete, Melshew Fenta Misker, Gebrehiwot Lema Legese.

**Data curation:** Assefa Belay Asrie.

**Formal analysis:** Assefa Belay Asrie.

**Investigation:** Habtamu Semagne Ayele, Kidist Goshime Tekle, Yonas Zewdu Milikit, Hiwot Tesfaselassie Afework, Gebrehiwot Lema Legese.

**Methodology:** Assefa Belay Asrie, Tafere Mulaw Belete, Melshew Fenta Misker, Gebrehiwot Lema Legese.

**Project administration:** Assefa Belay Asrie.

**Supervision:** Alemante Tafese Beyna, Habtamu Semagne Ayele, Kidist Goshime Tekle, Yonas Zewdu Milikit, Ephrem Adane Andargie, Hiwot Tesfaselassie Afework, Yenatfanta Gezu Lenjiso.

**Validation:** Alemante Tafese Beyna, Habtamu Semagne Ayele, Kidist Goshime Tekle, Yonas Zewdu Milikit, Ephrem Adane Andargie, Hiwot Tesfaselassie Afework, Yenatfanta Gezu Lenjiso.

**Writing – original draft:** Assefa Belay Asrie, Tafere Mulaw Belete, Melshew Fenta Misker, Alemante Tafese Beyna, Habtamu Semagne Ayele, Kidist Goshime Tekle, Yonas Zewdu Milikit, Ephrem Adane Andargie, Hiwot Tesfaselassie Afework, Yenatfanta Gezu Lenjiso, Gebrehiwot Lema Legese.

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
