## [Decision Letter · Decision Letter 0]

25 Jul 2025

Dear Dr. Assefa Belay Asrie,

Thank you for submitting your manuscript to PLOS ONE. After careful consideration, we feel that it has merit but does not fully meet PLOS ONE’s publication criteria as it currently stands. Therefore, we invite you to submit a revised version of the manuscript that addresses the points raised during the review process.

We recommend to address properly and in full all the changes/comments raised by reviewer.

We look forward to receiving your revised manuscript.

Kind regards,

Paolo Magni

Academic Editor

PLOS ONE

Journal Requirements:

2. In the online submission form you indicate that your data is not available for proprietary reasons and have provided a contact point for accessing this data. Please note that your current contact point is a co-author on this manuscript. According to our Data Policy, the contact point must not be an author on the manuscript and must be an institutional contact, ideally not an individual. Please revise your data statement to a non-author institutional point of contact, such as a data access or ethics committee, and send this to us via return email. Please also include contact information for the third party organization, and please include the full citation of where the data can be found.

Additional Editor Comments (if provided):

The paper requires some changes as indicated by reviewer 1. These requests need to be addressed in full and in an appropriate way.

Reviewers' comments:

Reviewer's Responses to Questions

**Comments to the Author**

1. Is the manuscript technically sound, and do the data support the conclusions?

Reviewer #1: Yes

2. Has the statistical analysis been performed appropriately and rigorously?

Reviewer #1: Yes

3. Have the authors made all data underlying the findings in their manuscript fully available?

Reviewer #1: Yes

4. Is the manuscript presented in an intelligible fashion and written in standard English?

Reviewer #1: Yes

Reviewer #1: This paper describes a cross-sectional cohort study of patients with type 2 diabetes attending a specialized hospital service in North West Ethiopia, and examines the relationship between a binary diabetes control category based on the average of three fasting blood glucose readings and health related quality of life. Overall the paper is technically sound and the findings are important in the context of a dearth of information from this region. The introduction is clear. The methods are fine, though it would be good to use SI units for the glucose measures. The sample size calculation seems convoluted – is this not just a convenience sample of patients, and if not, where did the figure of 62% come from? The results are clearly laid out and interpreted well in the discussion. Two minor points are that there is a typo in the title and the conclusions in the abstract are not necessarily justified by the findings described here.

**Do you want your identity to be public for this peer review?** For information about this choice, including consent withdrawal, please see our Privacy Policy

Reviewer #1: **Yes: ** Francis Finucane

---

## [Author Response · Author response to Decision Letter 1]

31 Jul 2025

Response to Editor and Reviewer Comments

Manuscript Title: Glycemic control and health-related quality of life and associated factors in type 2 diabetic patients attending a comprehensive specialized hospital in northwest Ethiopia

Manuscript Submission Number: PONE-D-25-24519

Journal: PLOS ONE

Response to the Editor

Dear Editor,

We extend our heartfelt gratitude to you for handling the publication process of our manuscript. We strongly believe that your skillful management and suggestions are instrumental in ensuring the smooth progress and successful completion of this endeavor.

Changes made or the new texts included in the revision process are highlighted in blue font color. Deletions are shown by red colored texts with Strikethrough line.

Journal Requirements:

We ensure that the manuscript meets the journal’s style requirements.

2. In the online submission form you indicate that your data is not available for proprietary reasons and have provided a contact point for accessing this data. Please note that your current contact point is a co-author on this manuscript. According to our Data Policy, the contact point must not be an author on the manuscript and must be an institutional contact, ideally not an individual. Please revise your data statement to a non-author institutional point of contact, such as a data access or ethics committee, and send this to us via return email. Please also include contact information for the third-party organization, and please include the full citation of where the data can be found.

Thank you, dear Editor, for your message. We would like to notify that our data is not associated with any proprietary restrictions. The statement provided in the online submission form regarding proprietary reasons was likely an unintentional error. Importantly, no such restriction was stated in the ‘Data availability statement’ of the originally submitted manuscript.

The full dataset is available in excel format and can be shared without restriction. It is currently held by the corresponding author and submitted as Supporting Information alongside the revised manuscript. The data availability statement has been revised accordingly.

The reviewer did not suggest any specific references for citation.

We have thoroughly reviewed the reference list to ensure it is complete and accurate. Additionally, we confirm that no retracted papers have been cited.

Additional Editor Comments (if provided):

The paper requires some changes as indicated by reviewer 1. These requests need to be addressed in full and in an appropriate way.

Thank you, dear Editor! We have addressed all the requests raised by Reviewer 1. A detailed point-by-point response is provided in the response letter, and the corresponding revisions are highlighted in the manuscript.

Response to Reviewer #1

Dear reviewer,

We are grateful for your time and dedication in reviewing our manuscript. Your insightful comments are greatly appreciated and have been used to help improve the quality of the manuscript. We have addressed the comments as follows.

Changes made or newly added text during the revision process are highlighted in blue font color. Deletions are shown by red colored texts with Strikethrough line.

Reviewer's Responses to Questions

Comments to the Author

1. Is the manuscript technically sound, and do the data support the conclusions?

Reviewer #1: Yes

Thank you!

2. Has the statistical analysis been performed appropriately and rigorously?

Reviewer #1: Yes

Thank you!

3. Have the authors made all data underlying the findings in their manuscript fully available?

Reviewer #1: Yes

Thank you!

4. Is the manuscript presented in an intelligible fashion and written in standard English?

Reviewer #1: Yes

Thank you!

5. Review Comments to the Author

Reviewer #1: This paper describes a cross-sectional cohort study of patients with type 2 diabetes attending a specialized hospital service in North West Ethiopia, and examines the relationship between a binary diabetes control category based on the average of three fasting blood glucose readings and health related quality of life. Overall the paper is technically sound and the findings are important in the context of a dearth of information from this region. The introduction is clear. The methods are fine, though it would be good to use SI units for the glucose measures. The sample size calculation seems convoluted – is this not just a convenience sample of patients, and if not, where did the figure of 62% come from? The results are clearly laid out and interpreted well in the discussion. Two minor points are that there is a typo in the title and the conclusions in the abstract are not necessarily justified by the findings described here.

Dear reviewer, we sincerely thank you again for your valuable feedback provided. We acknowledge the recognition of the technical soundness, clarity of the introduction, fineness of the methods, and clarity of the results and proper interpretation in the discussion. Moreover, we greatly acknowledge your recognition of the importance of the findings, in the context of limited information from the study region. We have addressed your comments as follows.

It would be good to use SI units for the glucose measures.

In accordance with this suggestion, we have revised the units for blood glucose measures in the manuscript to be presented in SI units (mmol/L). The following relation was used to change mg/dl values to mmol/L.

(mg/dl)/18=mmol/L

Clarification of the use of 62% in the sample size calculation

Thank you dear reviewer for your concern. We would like to clarify that this study is one of the themes of a broader project investigating the impact of herbal medicine use on clinical outcomes, including on blood glucose levels, and health-related quality of life among patients with type 2 diabetes. The sample size was, in fact, calculated using a single population proportion formula rather than through a convenience sampling method. The 62% figure used in the sample size calculation was based on the prevalence of herbal medicine use among type 2 diabetic patients, as reported in a previous study conducted in a similar setting. This proportion was chosen as the most appropriate for determining the required sample size for the project. We have already tried to state in the original submission why the figure (62%) was used in sample size calculation (with reference citation), in the methods section under subtitle “Sample size and sampling technique” just below the calculation of the sample size, page 5-6.

A typo in the title

Thank you for pointing out the typo in the title. We have corrected it, and the title is modified to: 'Assessment of glycemic control, health-related quality of life, and associated factors in type 2 diabetic patients attending a comprehensive specialized hospital in Northwest Ethiopia.' Additional changes made (other than typo correction) are intended to enhance the clarity of the title.

The conclusions in the abstract are not necessarily justified by the findings described here.

Upon consideration of this comment, we feel that there are statements which are not directly justifiable by the findings. With this note, we have deleted the statements. Please also note that we have made additional changes to this section based on the main results and with attention to enhancing clarity.

We have also made some changes to the conclusion of the main text (on page 26).

With sincere regards!

Assefa Belay Asrie (Corresponding author)

---

## [Editor Report · Decision Letter 1]

6 Aug 2025

Dear Dr.  Asrie,

We look forward to receiving your revised manuscript.

Kind regards,

Paolo Magni

Academic Editor

PLOS ONE

Journal Requirements:

Additional Editor Comments:

All reviewer’s comments should be addressed in full. Please use SI units for the glucose measures. The sample size calculation should be revised and better explained.

---

## [Author Response · Author response to Decision Letter 2]

8 Aug 2025

Response to Editor Comments

Manuscript Title: Assessment of glycemic control, health-related quality of life, and associated factors in type 2 diabetic patients attending a comprehensive specialized hospital in northwest Ethiopia

Manuscript Submission Number: PONE-D-25-24519R1

Journal: PLOS ONE

Dear Editor,

We sincerely thank you for your feedback. Your feedback is a valuable input to our efforts to improve the clarity of the sample size calculation and ensure consistency in reporting blood glucose measurements using appropriate units. Moreover, we are so much grateful for your prompt decisions, which are crucial to facilitate a timely final decision on our manuscript.

In response to your comments, we have made some additional revisions. This submission includes previous revisions plus some additional amendments on the sample size calculation section.

Journal Requirements:

No specific previously published work was recommended for citation.

We have checked our reference list. All references are available online and none of them are retracted.

Additional Editor Comments:

All reviewer’s comments should be addressed in full. Please use SI units for the glucose measures. The sample size calculation should be revised and better explained.

The paper reports interesting data.

Thank you dear Editor for recognizing that our paper presents interesting data.

SI units for glucose measurements:

All blood glucose values (which were expressed in mg/dl) have been converted to SI compatible, that is, millimoles per liter (mmol/L) throughout the manuscript.

These changes have been incorporated in the revised manuscript, and all modifications are highlighted in blue text for ease of visibility.

Although the actual SI unit for blood glucose measurement is mol/m³, glucose concentrations are commonly expressed in millimoles per liter (mmol/L) in biomedical and clinical practices. This unit is widely used in guidelines such as those of the American Diabetes Association, scientific literatures, and books. In response to the reviewer’s comment, we have changed all glucose concentration expressions in the manuscript from mg/dl to mmol/L. Even though mmol/L is not SI unit, it is an SI-compatible derived unit and is equivalent to mol/m³ and used commonly as alternative of mg/dL expressions practically. Mathematically, mmol/L is equal to mol/m³. (10-3×mol)/10-3m3 = mol/m3

Moreover, we would be happy if you could kindly specify the preferred SI unit you would like us to use, so we can make the necessary corrections accordingly.

References (just to support our reflections)

1. American Diabetes Association Professional Practice Committee (2024). 6. Glycemic Goals and Hypoglycemia: Standards of Care in Diabetes-2024. Diabetes care, 47(Suppl 1), S111–S125. https://doi.org/10.2337/dc24-S006

2. Renard E, Farret A, Kropff J, et al. Day-and-Night Closed-Loop Glucose Control in Patients With Type 1 Diabetes Under Free-Living Conditions: Results of a Single-Arm 1-Month Experience Compared With a Previously Reported Feasibility Study of Evening and Night at Home. Diabetes Care. 2016;39(7):1151-1160. doi:10.2337/dc16-0008

3. Baron DN, Broughton PM, Cohen M, Lansley TS, Lewis SM, Shinton NK. The use of SI units in reporting results obtained in hospital laboratories. Ann Clin Biochem. 1974;11(5):194-202. doi:10.1177/000456327401100154

The sample size calculation should be revised and better explained.

Dear Editor, based on your and the reviewer’s suggestions, we understand that the main clarity issue regarding the sample size calculation stems from the use of 62% (0.62) in our sample size calculation. We revised the section to provide a better explanation why this proportion was used.

We indicated that the sample size was determined using a single population proportion formula which is commonly used in population-based survey studies.

The 62% used in the calculation represents the prevalence of herbal medicine use among type 2 diabetes patients from a previous study in a similar setting. This study was conducted as one of the themes of a broader project proposed to assess the impacts of herbal medicine use on clinical outcomes and health-related quality of life among type 2 diabetes patients. In fact, herbal medicine use is considered as one of the factors associated with both glycemic control and health-related quality of life in this specific study. At the time of methodology development, we decided to base the sample size calculation on the prevalence of herbal medicine use reported in existing literature. Because the overall project also aimed to assess herbal medicine use. Since this variable was relevant across all themes of the broader study, including this specific manuscript, we considered it appropriate and applicable parameter for calculating the sample size for the project.

That is why we put this approach in the sample size calculation and we explained it subsequent to the formula. We have already attempted to explain this concept in the previous versions of the manuscript, and in this revision, we have worked on it to refine the clarity in brief and concise manner (on page 6 in the revised manuscript).

In summary:

• We used single population formula involving 62% and we get 362 initial sample size.

• We added 15% contingency for potential nonresponse and the final sample size considered for the study was 416.

• 386 were included in the final analysis (specific for this manuscript) because some participants were excluded essentially for data inconsistency problem.

With best regards,

Assefa Belay Asrie (Corresponding author)

---

## [Editor Report · Decision Letter 2]

2 Sep 2025

Assessment of glycemic control, health-related quality of life, and associated factors in type 2 diabetic patients attending a comprehensive specialized hospital in Northwest Ethiopia

PONE-D-25-24519R2

Dear Dr. Assefa Belay Asrie,

We’re pleased to inform you that your manuscript has been judged scientifically suitable for publication and will be formally accepted for publication once it meets all outstanding technical requirements.

Kind regards,

Paolo Magni

Academic Editor

PLOS ONE
---

## [Editor Report · Acceptance letter]

PONE-D-25-24519R2

PLOS ONE

Dear Dr. Asrie,

I'm pleased to inform you that your manuscript has been deemed suitable for publication in PLOS ONE. Congratulations! Your manuscript is now being handed over to our production team.

Kind regards,

on behalf of

Prof. Paolo Magni

Academic Editor

PLOS ONE